# Emerging Trends in Pathophysiology: Insights from the 9th International Congress of the International Society for Pathophysiology (ISP 2023)

**Anatolii Kubyshkin** [1,*], **Sergey Bolevich** [2], **Leonid Churilov** [3], **Vladimir Jakovljevic** [2,4], **Evgeniia Kovalenko** [1] **and Aleksandr Korovin** [3]

1    Department of General and Clinical Pathophysiology, V.I. Vernadsky Crimean Federal University, Vernadsky Avenue 4, Simferopol 295007, Russia; kovalenko_ep@mail.ru
2    Department of Human Pathology, First Sechenov University, Trubetskaia St. 8, Bld. 2, Moscow 119048, Russia; bolevich2011@yandex.ru (S.B.); drvladakgbg@yahoo.com (V.J.)
3    Department of Pathology, Saint Petersburg State University, Bld. 7-9, Universitetskaya Embk, Saint Petersburg 199034, Russia; l.churilov@spbu.ru (L.C.); korsyrik@mail.ru (A.K.)
4    Department of Physiology, University of Kragujevac, 34000 Kragujevac, Serbia
*    Correspondence: kubyshkin_av@mail.ru

**Abstract:** This article provides a summary of the 9th International Congress of the International Society for Pathophysiology (ISP 2023) which took place in Belgrade, Serbia, from 4 to 6 July 2023. It describes the main trends and the most promising areas of research in current pathophysiology, including investigations of new pathogenic pathways, and the identification of cellular and molecular mechanisms, target molecules, genetic mechanisms, and new therapeutic strategies. The present article also highlights the research conducted by leading scientific teams and national pathophysiological societies from various countries that adds to our understanding of the pathogenesis of diseases and pathological processes.

**Keywords:** pathophysiology; pathogenesis; molecular mechanisms

From 4 July to 6 July 2023, the 9th International Congress of Pathophysiology (ISP 2023) was held in Belgrade, Serbia. It should be noted that initially, it had been planned that the congress would take place in 2022 in Moscow on the campus of Sechenov University, but first the COVID-19 pandemic, and then the increasingly complicated international situation disrupted these plans. By the decision of the Council of the International Society for Pathophysiology, the congress was moved to Belgrade, and this site simultaneously hosted the 5th National Congress of Physiological Sciences of Serbia. Congress President V. Jakovljevich and Co-Chairman Sergei Bolevich made every effort to carry out all organizational activities in a short time to ensure the success of the ISP 2023 congress in the new location.

Currently, the International Society for Pathophysiology (ISP) unites scientists and specialists from 53 countries. It is clear that the organizing committee was aware of the difficulties and risks when planning the congress, so it is particularly gratifying to note that in the end, representatives of the pathophysiological community from 22 countries took part in its work. The total number of delegates, as at the last congress in Bratislava, were more than 200 people. Moreover, the representation included pathophysiologists from almost all continents. As before, Europe was represented by the largest number of participants (15 countries), but pathophysiologists from the USA and Canada, Qatar and Saudi Arabia, Tajikistan, Japan, and New Zealand also attended. As a result, almost 150 scientific papers presented by participants were published in the collected congress materials [1], including abstracts of 76 oral and 72 poster presentations held during plenary sessions, 16 symposia, and 2 poster sessions held at the congress. The scientific program reflected

various directions. It was clear that the use of cellular, molecular, and genetic approaches by pathophysiologists in scientific research to decipher the mechanisms responsible for the emergence and development of various pathological processes and socially significant diseases is on the increase.

The first plenary lecture at the congress was presented by the ex-president of the ISP, Olga Pechanova (Slovakia), and was devoted to comparing the effectiveness of the use of targeted therapy and saturated polyphenolic products of natural origin in experimental metabolic syndromes. It was emphasized that both types of therapy used statins and polyphenolic products to reduce oxidative stress and normalize the lipid profile, but simvastatin, unlike polyphenols, does not affect NO synthetase activity.

In total, five plenary lectures took place at the congress. An interesting plenary lecture was presented by Professor N. Dhalla (Canada). The study examined the molecular mechanisms underlying cardiac ischemic damage and the significance of adrenergic dysregulation in the development of heart failure. The lecture substantiated the important role of the loss of adrenergic regulation in connection with a decrease in the sensitivity of $\beta$1-adrenergic receptors and discussed metabolic changes with an increase in the expression of regulatory Gi-proteins and a decrease in Gs-proteins.

The lecture by M. Kreft (Slovenia), which also stimulated great interest, was dedicated to the study of previously understudied mechanisms of damage to the energy metabolism of the brain and, in particular, astrocytes. Metabolic plasticity has been shown to be largely associated with astrocyte reactivation. It is noted that the physiological and pathological properties of astrocyte metabolic plasticity have translational potential in identifying new potential diagnostic biomarkers and new therapeutic targets to mitigate neurodegeneration and age-related brain dysfunctions, including Alzheimer's disease.

Another plenary lecture by L. Kirshenbaum (Canada) was devoted to studying the participation of cytokine mechanisms in the formation of doxorubicin-induced cardiomyopathy. The authors examined the role of tumor necrosis factor-$\alpha$ (TNF-$\alpha$) in doxorubicin (DOX)-associated cardiac dysfunction. The results showed that there is a new signaling axis that functionally links the cardiotoxic effects of DOX to the proteasomal degradation of TRAF2 (TNF receptor-associated factor 2). Disruption of the critical survival pathway TRAF2 by DOX has been shown to sensitize cardiac myocytes to TNF-$\alpha$-mediated necrotic cell death and DOX cardiotoxicity.

The last plenary lecture, given by the author team of the congress hosts and presented by D. Djurich (Serbia), was devoted to studying the role of vitamins associated with homocysteine metabolism. It has been shown that hyperhomocysteinemia can damage the endothelium and walls of blood vessels, worsening the atherosclerotic process, with a negative impact on the mechanisms underlying myocardial infarction and heart failure, such as oxidative stress, inflammation, and changes in gasotransmitter function. The use of vitamin B6, which normalizes homocysteine metabolism, led to a decrease in oxidative stress and inflammation, the normalization of gasotransmitter function, and improvements in vasodilation and coronary blood flow in animal models.

In addition to plenary lectures, within the framework of the congress, as mentioned above, 16 symposia and 2 poster sessions were held on current problems in pathophysiology, which for the most part were moderated by representatives of various national societies in pathophysiology.

In the wake of one of the most pressing problems of 2020–2022—the new coronavirus infection—two symposia were held: "COVID-19—Epidemiological Alert—Where are we now?" and "COVID-19—Learning from Experience." Within the framework of symposia dedicated to the study of COVID-19, the report by L. Buryachkovskaya (Russia) was dedicated to the study of platelet function disorders during coronavirus infection and the impact of identified disorders on the course of post-COVID-19 syndrome. We also noted the report by L. Churilov (Russia) which studied the influence of antigenic mimicry of coronavirus proteins and human autoantigens in post-COVID-19 autoimmune lesions of the endocrine glands. In the section devoted to inflammation and immunity "Immunity &

Inflammation—From Basic Research to Clinical Application," the report by A. Kubyshkin and I. Fomochkina (Russia) was devoted to the study of the role of nonspecific proteinases and their inhibitors in systemic and local pathological processes. Their proposed classification of reactions in the proteinase inhibitor system at the systemic and local levels, which presented a classification of shock conditions based on the characteristics of the development of the systemic inflammatory response syndrome, attracted great attention.

At this congress, particular attention was paid to scientific research on heart pathology and diseases of the nervous system. Interesting reports were presented at the symposia on the study of cardiovascular protection and therapy (co-chairs Taskin Guven E. (Turkey) and Tipparaju S. (USA)); new pathogenic mechanisms of the development of cardiovascular pathology (co-chairs Lionetti V. (Italy) and Pierce G.N. (Canada)); the study of atypical connections and influences in cardiac diseases (co-chairs Turan B. (Turkey) and Zivkovic V. (Serbia)); and the molecular mechanisms of the development of neurological disorders (co-chairs Churilov L. (Russia) and Selakovic D. (Serbia)). Within the framework of the last symposium, the report by L. Churilov and co-authors (St. Petersburg) was devoted to the study of the role of endocrine factors in the development of autism.

Traditionally, much attention has been focused on the study of the molecular mechanisms of metabolic disorders, which was the subject of two symposia. At the first (co-chairs Bosch S. (France) and Falcon-Perez J.M. (Spain)), the role of extracellular vesicles in the development of metabolic disorders was discussed; at the second (co-chairs Buttar H.S. (Canada) and Tyagi S.C. (USA)), the possibility of using healthy nutritional strategies to prevent the development of metabolic diseases and improve quality of life was discussed. Special symposia were devoted to the study of oxidative stress (co-chairs Aburel O. (Romania) and Moskovtsev A. (Russia)), the role of mitochondria in the pathogenesis of diseases and the possibilities of targeted therapy (co-chairs Muntean D. (Romania) and Teixeira J. (Portugal)), and the molecular mechanisms of pathophysiological cascades in pathogenesis (co-chairs Todorovic Z. (Serbia) and Bozorgnia M. (Slovakia)). The congress did not ignore the discussion of issues of carcinogenesis (co-chairs Mogilenskikh A. (Russia) and Kreft M. (Slovenia)), the mechanisms of initiation and progression of tumors, as well as the search for new targets for their targeted therapy (co-chairmen Kukreja R. (USA) and Lukina S. (Russia)), and the problems of regenerative medicine (co-chairs Bradic J. (Serbia) and Desyatova M. (Russia)).

Within the framework of the congress, a symposium on the issues involved in improving teaching pathophysiology at medical universities was organized. As part of the discussion, Artem Grigoryan (Armenia) outlined an interesting approach that is increasingly used worldwide to describe the general mechanisms of disease development, emphasizing the so-called 'hallmarks'. Taken from the article by Hanahan D. and Weinberg R.A., The Hallmarks of Cancer [2], this term is increasingly used to describe the general mechanisms and distinctive features characteristic to the formation of various diseases. In particular, hallmark versions of descriptions are presented for the aging process [3], pulmonary hypertension [4], neurodegenerative diseases [5], and a number of other forms of pathology; an attempt has even been made to describe the general signs of health in a similar way [6].

Extensive poster sessions led by a large international team of moderators (Pierce G.N. (Canada); Dawn B. (USA); Kolesnik S. and Svetlikov A. (Russia); Selakovic D., Bradic J., Stojanovic A., and Joksimovic-Jovic J. (Serbia)) covered the entire range of pathophysiological studies from aerospace medicine to dentistry.

In conclusion, the congress in Belgrade resolved a number of organizational issues related to the workings of the ISP. It was noted that after the last congress in 2018 in Bratislava, the society encountered some difficulties. An updated website of the society was created (https://ispweb.cc, accessed on 18 January 2024), but as it is not updated regularly, many issues are not covered. The society's journal *Pathophysiology* (https://www.mdpi.com/journal/pathophysiology, accessed on 18 January 2024), whose Editor-in-Chief is Steven Alexander (USA), has resumed work on the MDPI platform, but so far, the journal

is in the third quartile (Q2). Soon after the congress, it became known that the journal *Pathophysiology*, part of the Scopus database, began to be indexed in the WoS database.

By tradition, the new president of the society is a representative of the host country of the ISP forum. This is the main organizer of the 9th ISP Congress, Professor Vladimir Jakovljevich (Serbia); we have great hopes that he will be able to maintain the stable and effective work of the ISP. At the same time, the Council of the ISP greatly appreciated the work of the organizing committee headed by the council in preparing the congress in Belgrade.

At the meetings of the ISP Council, ways to improve the activities of the society were discussed and the council of the society was updated.

The following were marked as the main priorities to be addressed by the new president and the members of the council:

- Strengthening and consolidating the International Society for Pathophysiology;
- Improving the ISP website;
- Activating the work of national societies on pathophysiology and stimulating individual membership;
- Strengthening the financial base of the society;
- Attracting young scientists to work in the society by holding regional scientific schools, training, and conferences for young researchers.

It was decided to hold the anniversary 10th Congress of the ISP in 4 years in Turkey in either 2026 or 2027. Moscow is being considered for the host of the 11th congress on the campus of the Sechenov University.

Today, pathophysiology as a science and as an academic discipline continues to develop all over the world and remains, under various names, one of the key basic disciplines in the training of doctors. Having outgrown the constraints of its traditional name, it performs the function of integrating systemic pathobiology with the medical sciences, providing the foundation for conducting scientific research in the field of intensively developing translational, precision, and restorative medicine.

**Author Contributions:** Conceptualization, A.K. (Anatolii Kubyshkin) and V.J.; analysis of reports, L.C. and A.K. (Aleksandr Korovin); writing—original draft preparation, A.K. (Anatolii Kubyshkin) and E.K.; writing—review and editing, A.K. (Anatolii Kubyshkin) and S.B.; supervision, A.K. (Anatolii Kubyshkin); project administration, V.J.; funding acquisition, L.C. and E.K. All authors have read and agreed to the published version of the manuscript.

**Funding:** Contribution of A.K. and L.C. supported by RF Grant NO. 075-15-2022-1110. Contribution of A.Ku. and E.K. supported by "Priority 2030" program and Grant FZEG-2023-0009.

**Institutional Review Board Statement:** Not applicable.

**Informed Consent Statement:** Not applicable.

**Data Availability Statement:** The raw data supporting the conclusions of this article will be made available by the authors on request.

**Conflicts of Interest:** The authors declare no conflicts of interest.

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
