# Peer review of "Emerging Trends in Pathophysiology: Insights from the 9th International Congress of the International Society for Pathophysiology (ISP 2023)"

_pathophysiology, doi:10.3390/pathophysiology31010011_

Round 1
Reviewer 1 Report
Comments and Suggestions for Authors
The manuscript includes a brief description of the presentations at the 9th International Congress of the International Society for Pathophysiology. The information is useful and interesting for researchers analyzing new trends in the mechanisms of pathogenesis of various diseases.
Reviewer 2 Report
Comments and Suggestions for Authors
This article describes the development of various new trends in pathophysiology as presented in a recent meeting. This article carefully describes the various sessions in the 9th International Congress of the International Society for Pathophysiology. The article is well written and very informative. I find little to criticize.
Reviewer 3 Report
Comments and Suggestions for Authors
This manuscript contains a summary of the 9th International Congress of the International Society for Pathophysiology (ISP 2023) celebrated earlier this year in Belgrade. The different topics addresses during the congress are clearly exposed, the challenges, the achievements and the difficulties of the ISP are addressed and priority actions are identified.
Minor points: from a formal point of view, note that sometimes the society is named as "International Society for Pathophysiology" even after the abbreviation ISP has been used. Equally, Presidium is written both with capital P or low-case p over the text. Please unify over the text.
